# Mapping the global design space of nanophotonic components using machine learning pattern recognition

Daniele Melati [1,4], Yuri Grinberg [2,4], Mohsen Kamandar Dezfouli[1], Siegfried Janz[1], Pavel Cheben[1], Jens H. Schmid [1], Alejandro Sánchez-Postigo [3] & Dan-Xia Xu [1*]

Nanophotonics finds ever broadening applications requiring complex components with many parameters to be simultaneously designed. Recent methodologies employing optimization algorithms commonly focus on a single performance objective, provide isolated designs, and do not describe how the design parameters influence the device behaviour. Here we propose and demonstrate a machine-learning-based approach to map and characterize the multi-parameter design space of nanophotonic components. Pattern recognition is used to reveal the relationship between an initial sparse set of optimized designs through a significant reduction in the number of characterizing parameters. This defines a design sub-space of lower dimensionality that can be mapped faster by orders of magnitude than the original design space. The behavior for multiple performance criteria is visualized, revealing the interplay of the design parameters, highlighting performance and structural limitations, and inspiring new design ideas. This global perspective on high-dimensional design problems represents a major shift in modern nanophotonic design and provides a powerful tool to explore complexity in next-generation devices.

[1] Advanced Electronics and Photonics Research Centre, National Research Council Canada, 1200 Montreal Rd., Ottawa, ON K1A 0R6, Canada. [2] Digital Technologies Research Centre, National Research Council Canada, 1200 Montreal Rd., Ottawa, ON K1A 0R6, Canada. [3] Universidad de Málaga, Departamento de Ingeniería de Comunicaciones, ETSI Telecomunicación, Campus de Teatinos s/n, 29071 Málaga, Spain. [4] These authors contributed equally: Daniele Melati, Yuri Grinberg. *email: dan-xia.xu@nrc-cnrc.gc.ca

A multitude of parameters determine the performance of a photonic device, encompassing the optical properties of the constituent materials, structural geometry, and dimensions. Similarly, the choice of the best design to proceed to fabrication, integration, and system implementation needs to take into account many performance criteria. These not only include the primary functionality but also other metrics, such as insertion loss, the effect of temperature, and the influence on other system components (e.g., back reflections), to name a few. Susceptibility of manufacturing yield to the inherent variability of the fabrication processes is another important consideration.

Historically, conception of a new device relies on theoretical knowledge and physical intuition to identify the potential structure and design parameter range. The design parameter space is explored semi-analytically or numerically, and the relevant performance metrics are analyzed. This approach is constrained in scope by computational resources and limited to structures governed by only a few parameters and where the evaluation process can be decomposed into sequential steps. As the scope of nanophotonics broadens in complexity and application range[1,2], this conventional approach poses increasing challenges. For example, in devices employing metamaterials[3–6] or complex geometries generated by inverse design and topology optimization[7–12], not only the number of design parameters vastly increase but they are often strongly interdependent. Sequential optimization is no longer applicable, and simultaneous optimization of multiple parameters is required.

Optimization tools such as genetic algorithm[13–15], particle swarm[16,17], and gradient-based optimization[18–21] are now commonly used to search more efficiently for high-performance designs[22]. More recently, supervised machine-learning methods, such as the artificial neural network, have begun to enter the fray in speeding up the search process[23,24]. While all these approaches represent significant improvements to the design flow, they still suffer constitutive limitations: usually a single performance criterion is optimized; only a single or a handful of optimized designs are discovered; and the optimization process needs to be repeated for modified performance criteria. Furthermore, optimized designs in isolation reveal very little on the characteristics of the design space and the influence of the design parameters on the device performance. Consequently, careful balancing of different figures of merit becomes difficult. A global perspective on the design space of nanophotonic devices is presently missing.

We propose here a methodology based on machine-learning (ML) tools to map and characterize a multiparameter design space. As a first demonstration-of-concept, we analyze a vertical fiber grating coupler in the silicon-on-insulator (SOI) platform consisting of multiple segments whose dimensions need to be optimized simultaneously[17,21,25–29]. From an initial sparse set of optimized designs, unsupervised dimensionality reduction reveals a lower-dimensional design sub-space where good designs with high-performance reside. Since computational effort grows exponentially with dimensionality, this sub-space can be mapped faster by orders of magnitude than the original design space, enabling the efficient evaluation and visualization of an arbitrary number of performance criteria. The comprehensive characterization of the continuous region that includes all possible good design solutions highlights their performance as well as structural differences and limitations. This provides a clear understanding of the design space, making possible the discovery of superior designs based on the relative priorities for a particular application. Furthermore, this global perspective on the design space can be exploited to arrive at conclusive arguments as to whether certain features can be obtained with a given structure. With the considered grating geometry, an upper limit to the minimum feature size is identified. This design bottleneck revealed by

dimensionality reduction inspired a new grating structure that incorporates subwavelength metamaterial and allows a minimum feature size greater than 100 nm without compromising the device performance. The composite design space of refractive index and segment dimensions involved here can be equally well characterized using the same global mapping procedure, indicating the general applicability of our optimization approach to a wide range of high-dimensional design problems. To the best of our knowledge, this is the first time such a global perspective is obtained by leveraging unsupervised machine-learning techniques for high-dimensional design problems in nanophotonics.

## Results

**Strategy for characterizing a multiparameter design space.** In designing multiparameter devices, it is often difficult or impossible to obtain extensive information on device performance variation over the large parameter space due to constrains on computational resources. We tackle this problem by introducing the three-stage process illustrated in Fig. 1.

In the first stage, multiple iterations of an optimization algorithm are used to generate a sparse collection of different good designs, i.e., designs that optimize a primary performance criterion (Fig. 1a). Supervised ML techniques are exploited to speed up the search process by quickly providing promising design candidates as starting points for the optimizer (see the Methods section). In the second stage (Fig. 1b), dimensionality reduction is applied to analyze the relationship in the parameter space between these degenerate designs. The goal is to find a lower-dimensional sub-space where all good designs reside. This design sub-space is described by significantly fewer parameters compared with the original design space. In the last stage (Fig. 1c), we map the design sub-space by computing across it all required performance criteria and identifying a continuous region of good design solutions. Through this process, the sparse initial set of good designs efficiently leads to the identification and comprehensive characterization of the continuum of all good designs in the sub-space.

A vertical fiber grating coupler with five segments per period is taken as the study case. The considered grating structure is illustrated in Fig. 2. Although desirable, perfectly vertical emission makes the design a challenging problem due to the necessity to suppress the second-order diffraction that reflects back into the waveguide[26]. In a recent work by Watanabe et al.[17], a single optimized design was generated using particle swarm optimization, providing a good fiber–chip coupling efficiency and a fairly low level of back reflections. Each period of the grating consists of a pillar of 220 nm in height and an L-shaped section with a partial etch to 110 nm[17]. The multiple segments in each period need to be simultaneously optimized, and therefore provide a good target to demonstrate our machine-learning-based design approach.

**Discovery of a sparse collection of good designs.** In the first stage, an in-house optimizer launched from random starting points in the original parameter space is used to search for the initial sparse set of grating designs with state-of-the-art fiber coupling efficiency. A supervised machine-learning predictor is trained to determine the diffraction angle of these complex gratings without performing a first principles Bloch mode calculation. The predictor is used to rapidly screen out random start designs that do not radiate close to the vertical direction, thereby speeding up the search process by ~250% (see the Methods section). This algorithm achieves a wide coverage of the initial design space.

For the grating illustrated in Fig. 2, the dimensions $[L_1 \ldots L_5]$ define the five-dimensional design parameter space we explore in

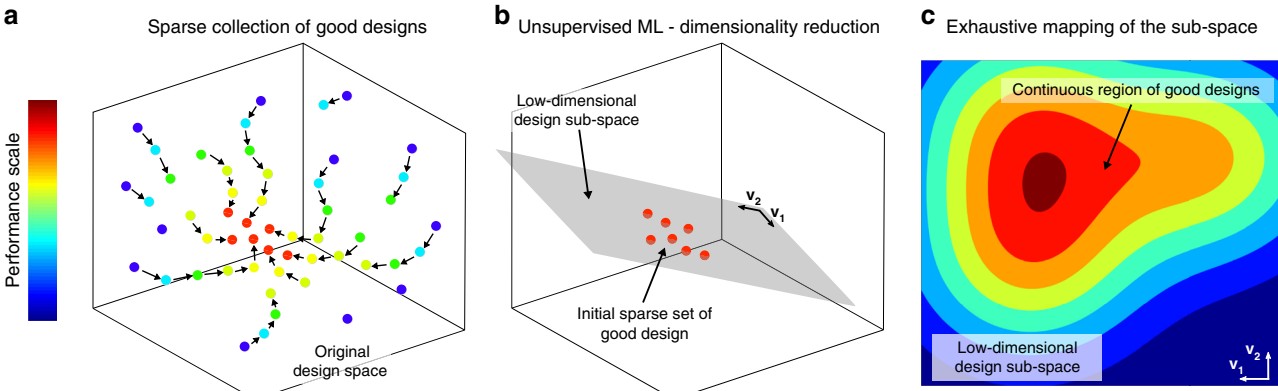

**Fig. 1** Mapping of a high-dimensional design space. The procedure in three stages is conceptually illustrated. **a** An initial sparse collection of good designs (red circles) is found by optimization (here, random re-start—as indicated by blue circles—followed by local search) in the original high-dimensional design space. A trained machine-learning predictor is used in conjunction with the optimizer to speed up the search by quickly identifying promising design candidates as starting points for the optimizer (see the Methods section). **b** Dimensionality reduction (e.g., principal component analysis) is employed to reveal the lower-dimensional sub-space where the good designs with high-performance reside, shown here as a hyperplane defined by the vectors $\mathbf{V_1}$ and $\mathbf{V_2}$. **c** The low-dimensional design sub-space can be exhaustively mapped. For the continuum of the designs in the sub-space (that includes also the initial sparse set), a complete characterization is made possible by computing both the performance criterion selected for optimization and any additional metric. Vectors $\mathbf{V_1}$ and $\mathbf{V_2}$ are the same as in **b** and are reported for clarity

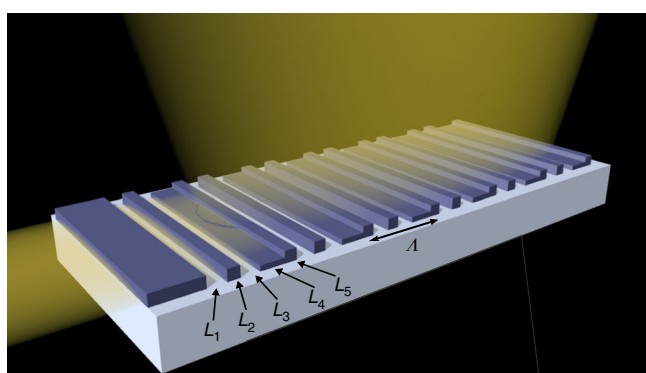

**Fig. 2** Schematic representation of the grating coupler structure. The guided light incident from the left is diffracted vertically by a grating periodically interleaving a pillar of height 220 nm and an L-shaped section partially etched to 110 nm. The L-shape approximates the angled facet of a conventional blazed grating[21] in a way that can be fabricated with standard lithography and etch methods. The pillars are designed to suppress back reflections by destructive interference. The five design parameters $L_1$–$L_5$ define the original five-dimensional parameter space and the grating period $\Lambda$

this work. As a primary optimization objective, we choose the coupling efficiency $\eta$ of the diffracted TE-polarized light to a standard single-mode optical fiber (SMF-28) placed vertically on top of the grating. Low back reflection $r$ is another important criterion in minimizing the impact of the grating to other upstream components in the system[30,31]. These considerations lead to the formulation of the optimization problem for the first stage represented in Fig. 1a as:

$$
\begin{aligned}
\underset{L_1 \cdots L_5}{\text{maximize}} \quad & \eta(L_1 \cdots L_5) \\
\text{subject to} \quad & r(L_1 \cdots L_5) < -15\,\text{dB} \\
& 400\,\text{nm} < \Lambda < 1\,\mu\text{m};\ L_i > 50\,\text{nm},\ i = 1 \cdots 5.
\end{aligned} \tag{1}
$$

The optimization is guided by a single performance metric, the coupling efficiency $\eta$. Only solutions with $\eta$ larger than 0.74 are retained, defined here as good designs. Back reflection $r$ is not

optimized, but simply constrained by rejecting design solutions with $r > -15$ dB. Additional constraints on the grating period $\Lambda$ and the minimum feature size are included to confine the optimization to designs that are physically manufacturable. The wavelength of light is set at $\lambda = 1550$ nm. A highly efficient Fourier-type 2-D eigenmode expansion simulator[32] is used to compute the device performance. As detailed in the next section, the optimization stage is halted after a sufficient number of good designs is collected. Each good design requires on average 1000 simulations to be identified (computational details in the Methods section).

**Sub-space identification through dimensionality reduction.** In this second and key step, we study the relationship between the sparse set of good solutions obtained solving the optimization problem (1) through machine-learning dimensionality reduction. The goal is to transform a set of correlated variables into a smaller set of new uncorrelated variables that retain most of the original information. Here, the linear principal component analysis (PCA)[33] is used obtaining a good level of accuracy (see PCA description in the Methods section).

We find that two principal components are sufficient to accurately represent the entire pool of good designs, each defined by five segment length in the original design space. That is, all good designs approximately lie on a 2D hyperplane—the reduced design sub-space. The rest of the design space can be excluded from further investigation. As we discuss in Supplementary Note 1, post-processing error analysis demonstrates that five good designs are sufficient for PCA to provide an accurate result. In order to verify convergence we collect here 45 good designs. The linear design sub-space is defined by two orthogonal basis vectors $\mathbf{V}_{1\alpha\beta}$ and $\mathbf{V}_{2\alpha\beta}$. Any design $k$ with dimensions $\mathbf{L}_k = [L_{1,k} \ldots L_{5,k}]$ can hence be written as

$$
\mathbf{L}_k = \alpha_k \mathbf{V}_{1\alpha\beta} + \beta_k \mathbf{V}_{2\alpha\beta} + \mathbf{C}_{\alpha\beta}. \tag{2}
$$

$\mathbf{C}_{\alpha\beta}$ is a constant vector that defines the reference origin on the hyperplane. Two scalar coefficients $\alpha_k$ and $\beta_k$ are thus sufficient to completely describe design $k$. Details of vector definitions are provided in the Methods section.

Now that the area of interest in the design space is limited to a 2D hyperplane, it becomes feasible to adopt a classical design

approach and perform an exhaustive mapping of this sub-space, as illustrated in Fig. 1c. First, we generate a uniform grid of 60 × 60 points covering the $\alpha$–$\beta$ hyperplane. Therefore, 3600 sampling points are sufficient to provide a wealth of information. As a comparison, sampling with the same resolution in the original design space would require ~1.5 million points, increasing the computation time by about over 400 times (details in the Methods section). For each point $[\alpha_k, \beta_k]$, we obtain the corresponding dimensions $[L_{1,k} \ldots L_{5,k}]$ in the original design space through Eq. (2) and compute the coupling efficiency $\eta$. The results are shown in Fig. 3a only for the designs with $\eta > 0.7$. Note that not all points on the $\alpha$–$\beta$ plane provide high coupling efficiency. A unit division in $\alpha$ or $\beta$ corresponds to a Manhattan distance $\sum_{i=1}^{5} |L_{i,A} - L_{i,B}|$ of 100 nm. The 60 × 60 grid covering the $\alpha$–$\beta$ hyperplane has a resolution of 5 nm in Manhattan

distance. This exhaustive mapping results in the discovery of a large and well-defined region of degenerate designs with $\eta > 0.74$, highlighted by the black contour line in Fig. 3a. This region encloses a continuum of designs in addition to those discovered in stage 1. Remarkably, although all the good designs have similar coupling efficiencies ranging from $\eta = 0.74$ to $\eta = 0.76$, the actual structure of the gratings can vary quite significantly, as will be detailed in the next section. Without dimensionality reduction, there is no obvious way to discover these alternative designs with similar coupling efficiency, but potentially different properties in other aspects.

As the last step, we validate the PCA outcome by verifying that the projection on the low-dimensional design sub-space (the $\alpha$–$\beta$ hyperplane) is sufficient to represent the region of good grating designs. We generate two additional 2D hyperplanes $\Gamma$–$\Pi$ and

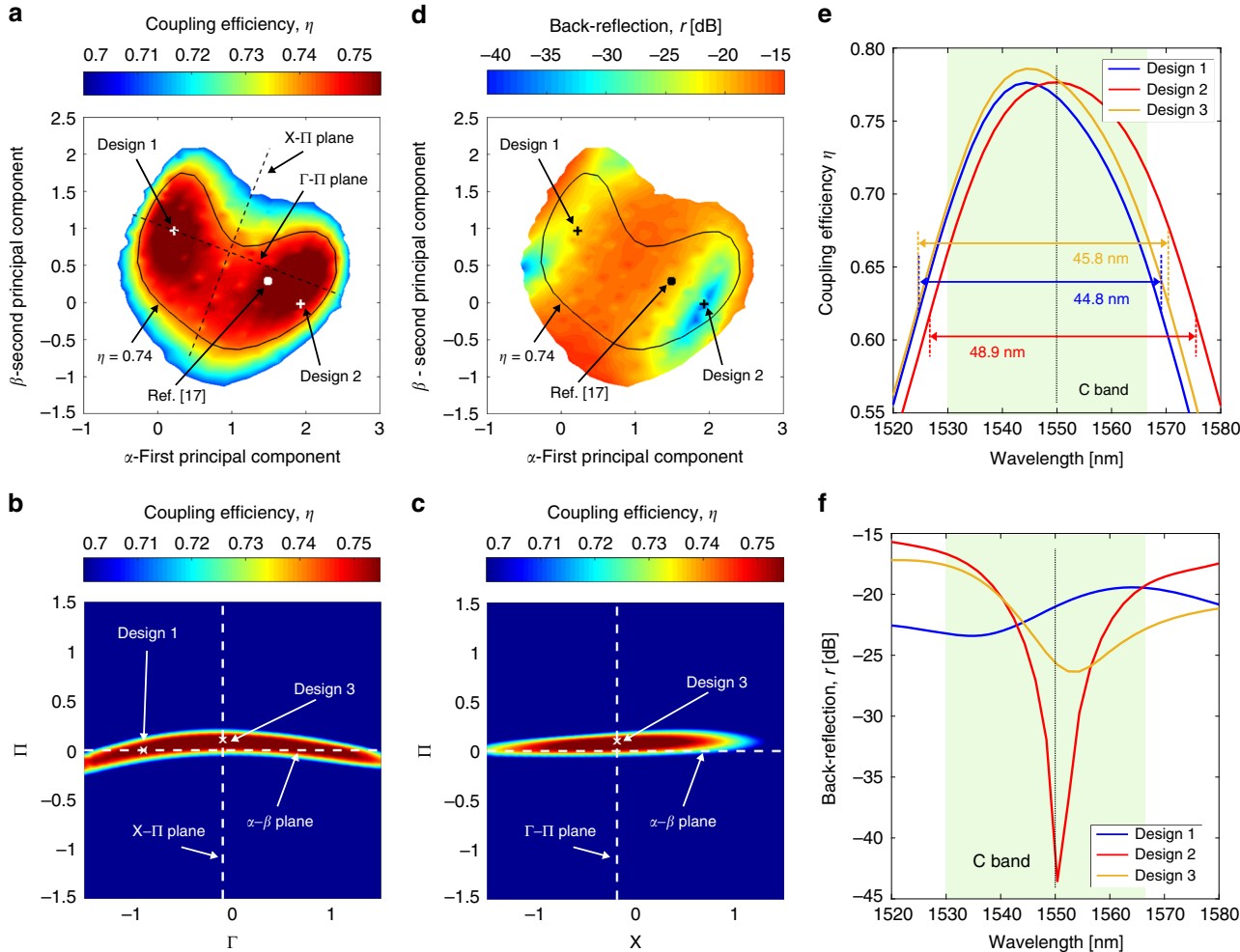

**Fig. 3** Exhaustive exploration of the lower-dimensional parameter sub-space. **a** Reducing the design parameters from the original five $L_i$ to the two principal component coefficients $\alpha$ and $\beta$ makes the exhaustive mapping of the sub-space of good designs achievable with modest computation resources. The map shows the coupling efficiency across the $\alpha$–$\beta$ hyperplane for $\eta > 0.70$. The large region of good designs with $\eta > 0.74$ is enclosed by the black contour line. Two designs with comparable coupling efficiency are marked along with the design reported in ref. [17]. **b, c** The coupling efficiency simulated across two 2-D hyperplanes ($\Gamma$–$\Pi$ and X–$\Pi$) orthogonal to the $\alpha$–$\beta$ hyperplane (whose intersections are shown with dashed white lines). $\Gamma$–$\Pi$ and X–$\Pi$ intersections with the $\alpha$–$\beta$ plane are shown in **a** with dashed black lines. Within these orthogonal planes, the cross-section of the sub-space of good designs reduces to a thin stripe confirming that it is approximately a 2-D geometrical structure. Design 3 represents the global optimum in both $\Gamma$–$\Pi$ and X–$\Pi$ projections: It has a coupling efficiency of 0.77, ~0.5% better than the top designs in the $\alpha$–$\beta$ hyperplane. The detailed structural parameters are reported in Table 1. **d** The back reflection $r$ simulated across the $\alpha$–$\beta$ hyperplane. **e, f** 2D finite-difference-time-domain (FDTD) simulations of **e** coupling efficiency and **f** back reflection as a function of wavelength for designs 1–3. The values obtained by FDTD are slightly different than that reported in the maps, but showing consistent trends. All three designs have a 1-dB bandwidth exceeding the telecommunication C band (1530 nm–1565 nm, green shaded area). Design 2 affords very low back reflections near 1550 nm, but only within a narrow wavelength band. In contrast, the back reflections of design 1 and 3 are less dependent on wavelength, but back reflection lower than −26 dB cannot be achieved

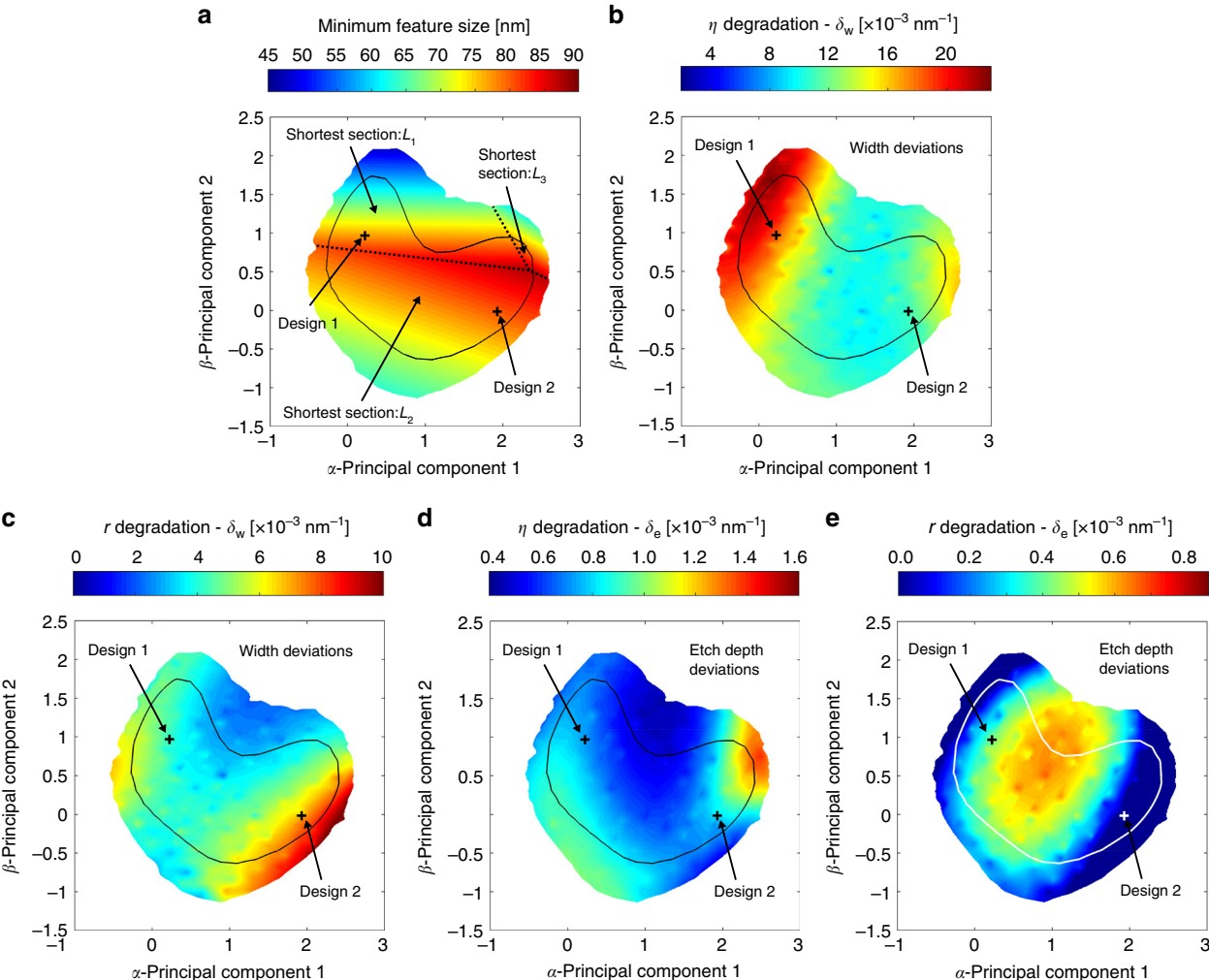

**Fig. 4** Comprehensive device characterization. The $\alpha$–$\beta$ hyperplane is exploited to investigate additional performance criteria. To ease comparison with Fig. 3, in each map, the solid contour encloses the region of good designs ($\eta > 0.74$) while the two crosses mark design 1 and 2. **a** Minimum feature size for the different designs. The map highlights also for which segment this minimum size occurs, identifying three areas (dashed black lines), where $L_1$, $L_2$, or $L_3$ is the shortest feature. Grating designs with minimum feature size greater than 88 nm do not exist. For almost the entire region of good designs, the minimum feature is either $L_1$ (e.g., for design 1) or $L_2$ (e.g., for design 2). **b-e** Sensitivity of the designs to **b**, **c** width deviations $\delta_w$ of both shallow and deeply etched sections in the grating and **d**, **e** etch depth deviations $\delta_e$ for the shallow-etched section. The maps show the value of the degradation derivative for **b**, **d** coupling efficiency $\eta$ and **c**, **e** back reflections $r$ across the sub-space of good designs as defined in the Methods section. Coupling efficiency of design 1 is more sensitive to width deviations compared with design 2. The opposite occurs for back reflections, where design 2 has a higher sensitivity than design 1. Both coupling efficiency and back reflection have a low sensitivity to etch depth variations within most of the region of good designs

X–$\Pi$ that are orthogonal to each other and to the $\alpha$–$\beta$ plane (details in the Methods section). They provide two different cuts through the $\alpha$–$\beta$ sub-space and their projections are shown in Fig. 3a with dashed black lines. We generate a uniform grid on $\Gamma$–$\Pi$ and X–$\Pi$ and simulate the coupling efficiency of the corresponding grating design. Coupling efficiencies $\eta > 0.7$ are plotted in Fig. 3b and c, respectively. The intersection with the $\alpha$–$\beta$ hyperplane is marked with a white dashed line. The axis uses the same scale as in Fig. 3a: a unit division on X, $\Gamma$, or $\Pi$ corresponds to a Manhattan distance of 100 nm. When projected on $\Gamma$–$\Pi$ and X–$\Pi$ hyperplanes, the region of good designs essentially reduces to a thin stripe whose thickness depends on the range of accepted coupling efficiencies $\eta$. This confirms that this region has approximately a 2D geometry. Although it appears slightly curved (see the $\Gamma$–$\Pi$ projection, Fig. 3b), it is still well approximated by the $\alpha$–$\beta$ hyperplane which has the advantage of being a simple linear structure.

**Characterization of the low-dimensional good design sub-space.** The exhaustive mapping of the sub-space of all good designs can now be readily extended to other performance metrics beyond the primary criterion originally chosen as the optimization objective (the coupling efficiency). This provides the designer a complete picture of the device behavior, including the upper and lower limit of each performance metric. Informed trade-offs and identification of the best design that fits specific application needs are hence made possible.

Along with the coupling efficiency, we evaluate here three additional criteria throughout the sub-space, i.e., back reflections, minimum feature size, and tolerance to fabrication uncertainty, as presented in Figs. 3 and 4. All maps use the same axis scale, range, and sampling as the $\alpha$–$\beta$ plane, and the black contour line marks the region with $\eta > 0.74$ for reference. Three designs are selected for further examination. Their structural and performance parameters are listed in Table 1. Designs 1 and 2 are on

**Table 1 Structural and performance parameters of selected grating designs as marked in Fig. 3**

| Design | $[\alpha, \beta]$ | $[L_1 \ldots L_5]$ [nm] | $\Lambda$ [nm] | Distance [nm] | $\eta$ | $r$ [dB] | BW [nm] |
|---|---|---|---|---|---|---|---|
| 1 | [0.22, 0.97] | [77, 84, 115, 249, 171] | 696 | – | 0.76 | −21 | 44.8 |
| 2 | [1.93, −0.02] | [102, 80, 117, 330, 98] | 727 | 216 | 0.76 | −37 | 48.9 |
| 3 | [0.97, 0.68][a] | [82, 87, 111, 283, 139] | 702 | 78 | 0.77 | −20 | 45.8 |
| ref. [17] | [1.49, 0.29] | [95, 83, 112, 314, 109] | 713 | 149 | 0.75[b] | −25[b] | 46.2[b] |

Distance refers to the Manhattan distance with respect to design 1. The coupling efficiency $\eta$ and reflection $r$ refer to the values at a wavelength of 1550 nm
[a]Closest projection on the hyperplane
[b]The performance for the structure proposed in ref. [17] is recalculated for consistency using the same Fourier-type 2D simulator and settings as the other structures

the $\alpha$–$\beta$ plane (marked on Fig. 3a, d), while design 3 (marked in Fig. 3b, c) is the global optimum in both $\Gamma$–$\Pi$ and $\mathrm{X}$–$\Pi$ projections (not exactly represented on the $\alpha$–$\beta$ plane). The design proposed in ref. [17], which was found through particle swarm optimization, also belongs to the sub-space of good solutions and its location on the $\alpha$–$\beta$ hyperplane is marked for reference. Despite the very different design parameter (especially for the L-shaped structure), all these gratings have a highly directional vertical emission and good overlap with the fiber mode, leading to a high coupling efficiency ($\eta > 0.75$ as listed in Table 1). On the other hand, the attainable back reflection differs markedly, from −21 dB for design 1 to −37 dB for design 2. The possibility to exhaustively map other metrics throughout the sub-space allows the designer to identify a design area with particularly low back reflections around design 2.

As an additional comparison of the performance for designs 1–3, Fig. 3e and f plots the two performance criteria $\eta$ and $r$ as a function of wavelength, now computed using 2D finite-difference-time-domain (FDTD) method as a cross-check. The results agree well with that predicted by the Fourier-type 2D simulator. All three designs have a 1-dB bandwidth larger than the telecommunication C band (1530 nm–1565 nm, green-shaded area) with design 2 slightly out-performing the other two. Regarding back reflections, the behavior of the three designs is remarkably different (Fig. 3f). Back reflections of design 2 near wavelength 1550 nm are very low, which is particularly important for coupling to a laser. However, a reflection of less than −30 dB can only be achieved within a 7-nm wavelength band. In contrast, the reflection of design 1 and 3 oscillates between −26 dB and −17 dB within the entire C band.

Minimum feature size determines the manufacturability of any nanophotonics device. Here, feature sizes can be easily retrieved exploiting the $\alpha$–$\beta$ hyperplane through a query process without performing any additional photonic simulations. For each point, the dimensions $[L_1 \ldots L_5]$ are computed with Eq. (2) and the shortest section is identified. Figure 4a shows the shortest segment among the five-dimensional parameters. It is immediately evident that a design with minimum feature size greater than 88 nm does not exist for this grating structure, even accepting a small penalty in the coupling efficiency. This essential information can be easily retrieved because of the global perspective that our method offers, and it would be difficult to obtain with a conventional optimization procedure. Clearly, the bottleneck is predominantly in either $L_1$ or $L_2$. This finding inspired an improved grating structure (described in the next section) that allows a minimum feature size greater than 100 nm while maintaining similar performance.

Another important aspect for nanophotonic devices is the robustness against unavoidable fabrication uncertainty. Here, we examine two sources of common dimensional variability: A width deviation $\delta_w$ for both shallow and deeply etched sections and etch depth deviation $\delta_e$ from the nominal 110 nm for the shallow-etched section. A good measure of the sensitivity of coupling

efficiency and back reflections to variability is provided by a quantity denoted here as a degradation derivative, defined as the average of the two directional derivatives with respect to positive and negative values of $\delta_w$ or $\delta_e$ (definition in the Methods section). The computed values of the four degradation derivatives are shown in Fig. 4b–e, with a high value indicating a high sensitivity. For width deviations, the coupling efficiency has a particularly sensitive region close to design 1. On the contrary, back reflection is more sensitive to width deviations in the region close to design 2. This region has a large overlap with the region of minimum back reflection shown in Fig. 3d, making design 2 and surrounding designs high-performing when back reflection is considered, but with stringent fabrication requirements. In Supplementary Note 2, we provide a direct verification of these results through a polynomial chaos-based stochastic analysis[22]. Regarding sensitivity to etch depth variations, the entire region of good designs largely overlaps with a region of low sensitivity for both coupling efficiency and back reflection.

**Dimensionality-reduction generality and ML inspired geometry.** The proposed design approach requires no physical assumptions on the device under study, enabling its application to other design problems with different types of input parameters and/or objectives. A straightforward demonstration of its generality is carried out by designing vertical grating couplers for the optical communication O-band, centered at 1310 nm (see Supplementary Note 4). We further demonstrate here this generality by investigating a new class of grating couplers utilizing subwavelength metamaterials[3,4] to achieve designs with a minimum feature size greater than 100 nm in both the propagation and the transverse directions. This new grating geometry (see Fig. 5) is inspired by the global mapping of minimum feature size described in the previous section. The optimization now involves both dimensions and the effective material index used to represent the subwavelength segments. Dimensions and refractive index have different numerical magnitudes, but they both significantly impact the device performance when varied. Below, we show that the method presented in the previous sections can successfully map out the high-performance region of this new mixed design space.

The subwavelength metamaterial grating structure, schematically shown in Fig. 5a, is represented by a mixed set of four geometrical parameters and one material parameter $\bar{\mathbf{L}} = \begin{bmatrix} L_1, L_2, L_3, L_4, n_{\mathrm{swg}} \end{bmatrix}$. These complex grating couplers can still be efficiently simulated using the 2D eigenmode expansion simulator. The formulation of the optimization problem remains the same as in the Eq. (1), but we additionally set $1.6 < n_{\mathrm{swg}} < 3$. Good designs found by the optimization algorithm ($\eta > 0.74$) are used to perform both PCA and the corresponding error analysis (see Supplementary Note 1). Since the new design space includes parameters of different nature, before executing PCA it is essential to normalize the variables through their statistically estimated standard deviations. Performing the PCA analysis

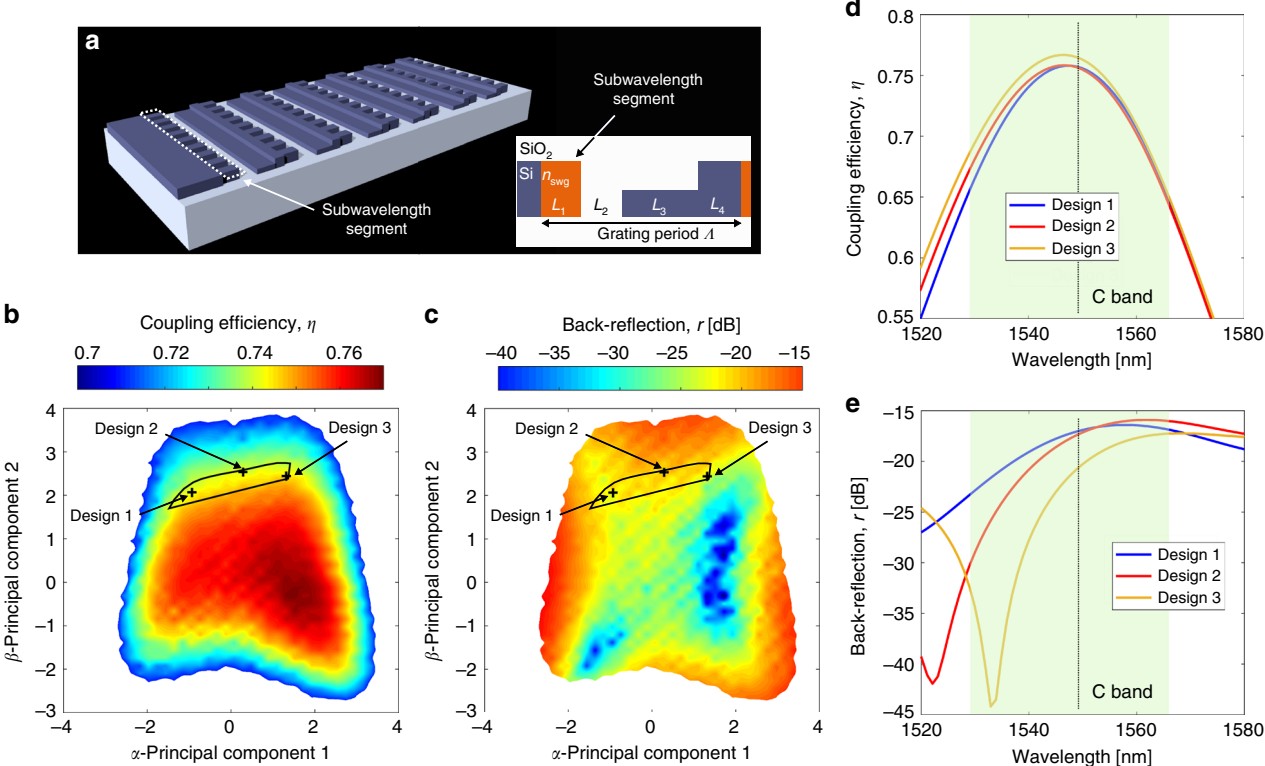

**Fig. 5** Vertical grating coupler with subwavelength metamaterial. **a** Schematic representation of the grating structure, with the subwavelength segment highlighted. The inset shows a 2D cross-section of the grating where the subwavelength metamaterial is modeled by an effective medium. Dimensionality reduction reveals that two principal components are sufficient to represent the good designs instead of the original five parameters. Plotted in **b** and **c** are the corresponding exhaustive maps of coupling efficiency and back reflection over the reduced parameters sub-space, respectively. Only designs with $\eta >$ 0.7 are shown. The black contour encloses the design region that ensures $\eta > 0.74$ (as in Fig. 3) and also a minimum feature size greater than 100 nm in both the propagation and transverse directions. These solutions were not possible with the grating structure shown in Fig. 2. **d**, **e** 2D-FDTD simulations of **d** coupling efficiency and **e** back reflection as a function of wavelength for designs 1–3. All three have $\eta > 0.74$ and back reflections below −15 dB within the C band

(stage 2) reveals that again only two principal components are sufficient to identify all the good designs within the original 5D design space (vectors are defined in the Methods section).

In Fig. 5b and c, we exhaustively map out the coupling efficiency and the back reflection over the 2D hyperplane (stage 3), discovering again a large continuous region with $\eta >$ 0.74. The map covers the region where the coupling efficiency exceeds 0.7. The black contour encloses all devices with a minimum feature size greater than 100 nm in both propagation and transverse directions and maintains $\eta \geq 0.74$. Three devices are marked on the maps for further investigation, where independent 2D-FDTD simulations provide full wavelength analysis in Fig. 5d, e (see Supplementary Note 3 for structural details and performance metrics). The results show that physically distinct devices with similar performances can be identified very efficiently. Delivering on such objectives would be very challenging by conventional optimization methods.

## Discussion

We have demonstrated a new approach for the design of complex photonics devices with a large number of parameters, case studied on a multiparameter vertical fiber grating coupler. Rather than generating a single optimized design solution, our methodology exploits the dimensionality reduction technique from the suite of machine-learning pattern recognition tools to identify within the large design space the lower-dimensional sub-space of good devices with high performance. This approach exponentially

scales down the complexity of the problem, making it feasible to exhaustively map the continuous region of grating couplers with comparable fiber coupling efficiencies ($\eta > 0.74$ at 1550 nm). Significant differences emerge when different performance criteria are considered, such as back reflection, minimum feature size, and tolerance to fabrication uncertainty. Such a global perspective also reveals performance and structural limitations of the design geometry. In particular, we were able to conclude that good coupling efficiency and a minimum feature size greater than 88 nm could not be obtained simultaneously for this first structure. The analysis of this shortcoming inspired a new class of grating couplers that uses subwavelength metamaterial, achieving a minimum feature size greater than 100 nm while maintaining state-of-the-art coupling efficiency and back reflection.

Given the generality of our implementation, the presented methodology can be readily exploited to navigate and comprehend a wide range of high-dimensional design spaces that photonic designers often encounter. While it is demonstrated here for two different design problems in nanophotonics, applications to photonic circuits or even sub-systems can be foreseen. This design methodology opens up new avenues in photonic device analysis and design where other dimensionality reduction methods such as Kernel PCA[34], Principal manifolds[35], and Auto-encoders[36] can deal with navigating through even more complex design spaces. Indeed, automation and integration of dimensionality reduction within the design flow will provide a powerful platform potentially transforming how advanced photonic devices are discovered and investigated.

## Methods

**Grating coupler simulation**. The simulation of coupling efficiency and back reflections for each design of the grating coupler is performed exploiting either a 2D vectorial Fourier eigenmode expansion simulator[32] or a commercial 2D-FDTD solver. We consider a structure including silicon substrate, 2-μm buried oxide, 220-nm-thick silicon core and a silica upper cladding of 1.5 μm thickness. Silicon and silica refractive indices are 3.45 and 1.45 at $\lambda = 1550$ nm. The mode of an SMF-28 single-mode optical fiber vertically coupled on top of the grating is modeled with a Gaussian function with a mode field diameter of 10.4 μm ($\lambda = 1550$ nm). The fiber facet is assumed to be in direct contact with the top of the upper cladding and its longitudinal position along the grating is optimized for each design to maximize the coupling efficiency. The latter is calculated as the overlap integral between the simulated field diffracted upwards by the grating and the Gaussian function.

**Machine learning enhanced optimization**. We implemented a random restart local-search algorithm to solve problem (1), although other search methods could also be used. For each initial random design, small perturbations are made until a better solution in terms of coupling efficiency is found and a line search is exploited to seek further improvement until convergence. The perturbation and line search process is repeated until no improvement is found in the perturbation stage.

Following an initial optimization round with random-restart where a small collection of good designs was obtained, we trained a supervised machine-learning model, specifically gradient boosted trees, to predict if the radiation angle is within 5° of vertical. Once trained, we used the predictor to sample random designs that are nearly vertical while rejecting other random designs. Only near vertical emitting designs proceed to the local search stage. The use of the predictor in the optimization led to ~250% increase in optimizer speed to find new designs meeting the coupling efficiency criteria $\eta > 0.74$.

The performance of the general purpose machine-learning predictor is comparable to that of a predictor based on the scalar grating equation relating the section lengths, effective indices and the radiation angle:

$$\sum_{i=1}^{5} N_i L_i = c + a \sin \theta \cdot \sum_{i=1}^{5} L_i,$$

where $a, c$ are constants related to the wavelength and the overcladding effective index, $N_i$ is the effective index of the $i$th section, $L_i$ is the length of the $i$th section, and $\theta$ is the radiation angle. Given the simulated data ($(\theta, L_i)$, one can use linear regression to estimate all the constants and then use those to predict the radiation angle of the structure for any combination of the section lengths. Despite the fact that the general machine-learning predictor is not aware of this approximation, its predictions are comparable with those obtained using the scalar grating equation. Furthermore, this approach can be applied to predict other quantities that cannot be described by simple closed-form expressions.

**PCA**. PCA is a dimensionality reduction technique that has been used widely and successfully across various engineering and science disciplines[37–40] and is implemented in most scientific computing platforms (e.g., Matlab, R, Python, etc.). Consider $m$ data points in an $n$-dimensional design space. This can be written as a centered data matrix $\mathbf{L} \in \mathbb{R}^{m \times n}$, where the statistical average along each dimension is subtracted. PCA finds a sequence of best orthogonal linear projections (called principal components) that maximizes the corresponding variances. Mathematically, this is done by finding a set of vectors $\mathbf{V}_1, \mathbf{V}_2, \dots, \mathbf{V}_n \in \mathbb{R}^n$ that are L2 normalized and orthogonal, $\|\mathbf{V}_i\|_2 = 1, \mathbf{V}_i \perp \mathbf{V}_j$, and for every $k < n$, minimize

$$\left\| \mathbf{LR} - \mathbf{L} \left[ \mathbf{R_k}, \mathbf{0}^{n \times (n-k)} \right] \right\|_2,$$

where, $\mathbf{R} = [\mathbf{V}_1, \mathbf{V}_2, \dots, \mathbf{V}_n] \in \mathbb{R}^{n \times n}$ is the transformation matrix that is formed by using the PCA vectors as its columns, $\mathbf{R_k} = [\mathbf{V}_1, \mathbf{V}_2, \dots, \mathbf{V}_k] \in \mathbb{R}^{n \times k}$ is similar but transforms to a lower-dimensional space consisting of only k first components, and $\mathbf{0}^{n \times (n-k)}$ is a null matrix used for padding. Such an exercise also returns the weights associated with PCA vectors from which the main principal components can be selected and the remaining can be dropped, with negligible amount of information lost. The original data set can now be approximately represented by $k < n$ effective parameters using a matrix product $\mathbf{L}^{\text{PCA}} = \mathbf{LR}_k$, where each row corresponds to a transformed representation of a single data point. The transformation back to the original $n$-dimensional space can be done using $\mathbf{L}^{\text{est}} = \mathbf{L} \, \mathbf{R}_k \mathbf{R}_k^{\mathrm{T}}$, which can be used to quantify approximation errors incurred by reducing the dimensionality.

**Hyperplanes definitions**. The hyperplane approximation given in Eq. (2) and computed by PCA is defined by three 5D vectors $\mathbf{V}_1$, $\mathbf{V}_2$, and C, the latter being the reference origin within the hyperplane. The scaled vectors computed for the grating structure in Fig. 2 are: $\mathbf{V}_{1\alpha\beta} = [-0.43, 3.78, -20.82, 44.77, -30.21]$ nm, $\mathbf{V}_{2\alpha\beta} = [-25.80, 10.81, -37.69, -3.86, 21.93]$ nm and the origin $\mathbf{C}_{\alpha\beta} = [102, 73, 156, 243, 156]$ nm.

For the two orthogonal hyperplanes Γ–Π and Χ–Π described in Fig. 3b and c, the vectors are found following linear algebra considerations. Γ–Π is defined enforcing the plane to pass through the two points (defined in the 5D design space) [77, 84,

115, 249, 171] nm (design 1 in Table 1) and [95, 83, 104, 336, 98] nm and to be orthogonal to the $\alpha$–$\beta$ hyperplane. The resulting vectors are $V_{1\Gamma\Pi} = [9.45, -0.35, -6.13, 45.95, -38.35]$ nm, $V_{2\Gamma\Pi} = [-22.48, 33.08, 9.87, -17.87, -28.78]$ nm, $\mathbf{C}_{\Gamma\Pi} = [85, 84, 110, 289, 138]$ nm. Χ–Π is defined as orthogonal to both $\alpha$–$\beta$ and Γ–Π and passing through the point [82, 87, 111, 283, 139] nm (design 3 in Table 1). The vectors are $V_{1\times\Pi} = [-25.92, 11.96, -44.16, 10.05, 12.61]$ nm, $V_{2\times\Pi} = V_{2\Gamma\Pi}$, and $\mathbf{C}_{\times\Pi} = [85, 84, 110, 284, 142]$ nm.

For the grating structure with subwavelength transverse metamaterial (Fig. 5a), the 5D vectors defining the 2D hyperplane are $\mathbf{V}_{1\alpha\beta} = [13.37 \text{ nm}, -3.34 \text{ nm}, 17.21 \text{ nm}, -19.5 \text{ nm}, -0.03]$, $\mathbf{V}_{2\alpha\beta} = [5.07 \text{ nm}, 14.07, -7.33 \text{ nm}, 2.53 \text{ nm}, -0.076]$ and $\mathbf{C}_{\alpha\beta} = [272 \text{ nm}, 71 \text{ nm}, 247 \text{ nm}, 120 \text{ nm}, 2.63]$.

**Computational resources**. In the proposed method, the largest fraction of the computational time is dominated by design simulations with a negligible overhead time for data processing. The initial optimization (stage 1) enhanced by the ML angle predictor required ~5000 photonic simulations to identify the initial five good designs used to find the reduced design sub-space through PCA. The exhaustive mapping was then performed by sampling 3600 points (design) arranged in a square grid in the sub-space and doing the corresponding 3600 photonic simulations to compute simultaneously coupling efficiency and back reflections. Other sampling strategies or sparser grids can be used to reduce the number of simulations. Likewise, a different simulation approach could be used to retrieve additional metrics within the same simulation. The computation of additional performance metrics may require additional simulations, for example, when fabrication tolerance is estimated through the degradation derivatives as reported in Fig. 4b–e. On the other hand, the computation of the minimum feature size does not require any additional simulation.

It is worth calculating the number of points of a grid with the same resolution used for the $\alpha$–$\beta$ hyperplane, but across the five grating dimensions and covering all designs with coupling efficiency $\eta > 0.74$ found in stage 1. The five dimensions of these good designs span a range of 60 nm, 27 nm, 86 nm, 138 nm, 118 nm, respectively. A grid with 5 -nm resolution across all segments and covering all good designs, including two extra points to confirm the boundaries per segment, results in $14 \times 7 \times 19 \times 30 \times 26 \approx 1.5 \cdot 10^6$ points (designs). Compared with mapping directly the original 5D design space, mapping the $\alpha$–$\beta$ hyperplane thus reduces the computation time of about 400 times. For higher dimensions, the reduction can be even more significant.

**Uncertainty model**. For the investigation of design tolerance to fabrication uncertainty, we assume a width deviation $\delta_w$ for both shallow-etched and deeply etched sections

$$L'_1 = L_1 - \delta_w; \; L'_2 = L_2 + \delta_w; \; L'_3 = L_3 - \delta_w; \; L'_4 = L_4; \; L'_5 = L_5 + \delta_w$$

We define a degradation derivative that is computed from two directional derivatives, assuming that over-etch and under-etch are equally likely (positive and negative values of $\delta_w$ can in general affect the device performance differently). Calculating the common derivative would not be informative, as for locally optimized devices it would be close to zero. For coupling efficiency, we are interested in calculating

$$\alpha_\eta = -\frac{1}{2} \left( \frac{\partial^+ \eta}{\partial \delta_w} - \frac{\partial^- \eta}{\partial \delta_w} \right) \cong -\frac{1}{2|\Delta\delta_w|} \left( \eta^+ + \eta^- - 2\eta_0 \right),$$

where $\partial^+$ and $\partial^-$ are the derivatives computed for positive and negative values of $\delta_w$. The minus sign ensures that a positive value of $\alpha_\eta$ indicates a worse (lower) coupling efficiency. Derivatives are numerically computed considering a small width variation and simulating the coupling efficiencies $\eta^+$ (when $\Delta\delta_w = 5$ nm) and $\eta^-$ (when $\Delta\delta_w = -5$ nm). $\eta_0$ is the coupling efficiency for $\delta_w = 0$. Similarly, for back reflections

$$\alpha_r = \frac{1}{2} \left( \frac{\partial^+ \eta}{\partial \delta_w} - \frac{\partial^- \eta}{\partial \delta_w} \right) \cong \frac{1}{2|\Delta\delta_w|} \left( \eta^+ + \eta^- - 2\eta_0 \right).$$

Also in this case, a positive $\alpha_r$ indicates worse (higher) back reflections. The degradation derivatives plotted in Figs. 4b and c are finally computed as

$$d = \begin{cases} \alpha & \text{if } \alpha > 0 \\ 0 & \text{otherwise} \end{cases}$$

When etch uncertainty is considered, $\delta_e$ represents the variability on the 110 nm etch depth in the fourth section of the grating. The same definitions apply for the degradation derivatives.

## Data availability
The data that support the plots within this paper and other findings of this study are available from the corresponding author upon request.

## Code availability
The custom code that has been used to generate the results reported in this paper is available from the corresponding author upon request.

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

## Author contributions

D.M., Y.G. and D.X.X. conceived the design approach and developed the theoretical framework. Y.G. developed the machine-learning algorithms. D.M. analysed the data and performed the stochastic analyses. P.C., A.S.P. and J.H.S. assisted in selecting the grating coupler study case. A.S.P. contributed to the development of the interface between the photonic simulator and machine-learning algorithms. S.J., P.C. and J.H.S. provided theoretical and design guidance. D.X.X. and Y.G. supervised the project. M.K.D. and A.S.P. conceived and analysed the grating design with subwavelength patterning. All authors contributed to the discussion and paper preparation.

## Competing interests

The authors declare no competing interests.
