## [Peer Review File · Nature Communications]

Editorial Note: This manuscript has been previously reviewed at another journal that is not operating a transparent peer review scheme. This document only contains reviewer comments and rebuttal letters for versions considered at Nature Communications.`

REVIEWERS' COMMENTS:

Reviewer #1 (Remarks to the Author):

The authors have made a significant and meaningful effort to address my concerns on their original submission. The revised paper makes a far more appealing and convincing case on the importance of the reported work. On top of all introduced clarifications, I find particularly insightful the newly presented example of an optimized grating coupler -based upon a sub-wavelength grating structure- that is designed following the relevant information and guidelines provided by the machine-learning analysis of the original device.

Following these changes, I am fully persuaded of the potential impact of the introduced ideas looking forward, so I think the paper should be a good fit for publication in Nature Communications.

This is high quality research work. Moreover, the manuscript is very well written, the material is well organized and the latest clarifications and additions are of great value. I have no further technical questions or any other suggestions on the paper, so my recommendation is to accept the paper in its present form.

Reviewer #3 (Remarks to the Author):

I read the manuscript and the long Response addressing all the comments provided by the Reviewers after the previous submission.

The manuscript has been amended following all the requests and comments raised by the reviewers and the Authors provided an exhaustive, long and motivated response. In addition to the response to the Reviewers the Authors provided a long initial motivation letter highlighting advantages, potentials as well as limitations of the proposed approach. Entire parts of the original manuscript have been rewritten and new parts have been included. I found particularly relevant the section on "Sub-space identification through dimensionality reduction" and "Generality of dimensionality reduction...". These have a broad validity making the manuscript adapt for Nat. Comms.

Personally, I'm fully satisfied by all the amendments and the modifications to the manuscript. The paper is now even more suitable for a broad audience and applicable to a wider class of problems. It now includes tolerance studies and other details required by the other Reviewers. I see the grating coupler problem considered in the manuscript "just" as a practical illustrative example to handle complex problems but the generality of the approach goes well beyond this structure. The paper is now organized providing the readers information and tools to apply the approach to other cases.

I do not have particular remarks on the scientific level of the manuscript that I consider suitable and recommendable for the publication in Nature Communications.